# Molecular diversity of *Giardia duodenalis* in children under 5 years from the Manhiça district, Southern Mozambique enrolled in a matched case-control study on the aetiology of diarrhoea

**Augusto Messa, Jr.**[1], **Pamela C. Köster**[2], **Marcelino Garrine**[1,3], **Carol Gilchrist**[4], **Luther A. Bartelt**[5], **Tacilta Nhampossa**[1,6], **Sérgio Massora**[1], **Karen Kotloff**[7], **Myron M. Levine**[7], **Pedro L. Alonso**[1,8], **David Carmena**[2]*, **Inácio Mandomando**[1,6]*

1 Centro de Investigação em Saúde de Manhiça (CISM), Maputo, Mozambique, 2 Parasitology Reference and Research Laboratory, National Centre for Microbiology, Health Institute Carlos III, Majadahonda, Madrid, Spain, 3 Global Health and Tropical Medicine (GHTM), Instituto de Higiene e Medicina Tropical (IHMT), Universidade Nova de Lisboa (UNL), Lisbon, Portugal, 4 University of Virginia, Charlottesville, Virginia, United States of America, 5 Division of Infectious Diseases, University of North Carolina at Chapel Hill, Chapel Hill, North Carolina, United States of America, 6 Instituto Nacional de Saúde (INS), Ministério da Saúde, Maputo, Mozambique, 7 Center for Vaccine Development (CVD), University of Maryland School of Medicine, Baltimore, Maryland, United States of America, 8 ISGlobal, Hospital Clínic-Universitat de Barcelona, Barcelona, Spain

☯ These authors contributed equally to this work.
* dacarmena@isciii.es (DC); inacio.mandomando@manhica.net (IM)

## Abstract

*Giardia duodenalis* is an enteric parasite commonly detected in children. Exposure to this organism may lead to asymptomatic or symptomatic infection. Additionally, early-life infections by this protozoan have been associated with impaired growth and cognitive function in poor resource settings. The Global Enteric Multicenter Study (GEMS) in Mozambique demonstrated that *G. duodenalis* was more frequent among controls than in diarrhoeal cases (≥3 loosing stools in the previous 24 hours). However, no molecular investigation was conducted to ascertain the molecular variability of the parasite. Therefore, we describe here the frequency and genetic diversity of *G. duodenalis* infections in children younger than five years of age with and without diarrhoea from the Manhiça district in southern Mozambique enrolled in the context of GEMS. Genomic DNA from 757 *G. duodenalis*-positive stool samples by immunoassay collected between 2007–2012, were reanalysed by multiplex PCR targeting the *E1-HP* and *C1-P21* genes for the differentiation of assemblages A and B. Overall, 47% (353) of the samples were successfully amplified in at least one locus. Assemblage B accounted for 90% (319/353) of all positives, followed by assemblage A (8%, 29/353) and mixed A+B infections (1%, 5/353). No association between the presence of a given assemblage and the occurrence of diarrhoea could be demonstrated. A total of 351 samples were further analysed by a multi-locus sequence genotyping (MLSG) approach at the glutamate dehydrogenase (*gdh*), ß-giardin (*bg*) and triose phosphate isomerase (*tpi*) genes. Overall, 63% (222/351) of samples were genotyped and/or sub-genotyped in at least

**Data Availability Statement:** All relevant data are within the manuscript and its Supporting Information files.

**Funding:** GEMS was funded by the Bill and Melinda Gates Foundation through the Center for Vaccine Development at the University of Maryland, School of Medicine who coordinated GEMS-1 and GEMS-1A, with grant number 38874 (MML) and OPP1033572 (MML), respectively. DC received funding from the Health Institute Carlos III, Ministry of Economy and Competitiveness (Spain), under project PI16CIII/00024.TN received funding from the Fundo Nacional de Investigacão, Ministry of Science and Technology (Mozambique) under project 245-INV. Additional funding was obtained from the USAID Country Office of Mozambique under the Fixed Amount Award Grant No. AID-656-F-16-00002 (IM). CISM is supported by the Government of Mozambique and the Spanish Agency for International Development Cooperation (AECID). The funders had no role in the study design, data collection and analysis, decision to publish, or preparation of the manuscript.

**Competing interests:** The authors have no competing interests.

one of the three markers. Sequence analysis revealed the presence of assemblages A (10%; 23/222) and B (90%; 199/222) with high molecular diversity at the nucleotide level within the latter; no mixed infections were identified under the MLSG scheme. Assemblage A sequences were assigned to sub-assemblages AI (0.5%, 1/222), AII (7%, 15/222) or ambiguous AII/AIII (3%, 7/222). Within assemblage B, sequences were assigned to sub-assemblages BIII (13%, 28/222), BIV (14%, 31/222) and ambiguous BIII/BIV (59%, 132/222). BIII/BIV sequences accumulated the majority of the single nucleotide polymorphisms detected, particularly in the form of double peaks at chromatogram inspection. This study demonstrated that the occurrence of gastrointestinal illness (diarrhoea) was not associated to a given genotype of *G. duodenalis* in Mozambican children younger than five years of age. The assemblage B of the parasite was responsible for nine out of ten infections detected in this paediatric population. The extremely high genetic diversity observed within assemblage B isolates was compatible with an hyperendemic epidemiological scenario where infections and reinfections were common. The obtained molecular data may be indicative of high coinfection rates by different *G. duodenalis* assemblages/sub-assemblages and/or genetic recombination events, although the exact contribution of both mechanisms to the genetic diversity of the parasite remains unknown.

## Author summary

*Giardia duodenalis* is a globally distributed enteric parasite, and the most detected intestinal protozoan in children. Infection by this pathogen can evolve from asymptomatic to chronic disease with a range of gastrointestinal manifestations including diarrhoea. In endemic areas, early infection in childhood can contribute to impaired growth and cognitive function. This organism is currently divided in 8 distinct (A-H) genetic groups (assemblages) with marked differences in host range and specificity and even geographical distribution. Therefore, molecular studies aiming at investigating the genetic diversity of the parasite are cornerstone to assess its transmission dynamics. By using targeted-molecular genotyping methods based on PCR and sequencing, we demonstrated here that i) the occurrence of diarrhoea in Mozambican young children infected with *G. duodenalis* was not linked to a given assemblage of the parasite, ii) assemblage B was the most prevalent genetic variant of *G. duodenalis* circulating in the surveyed paediatric population, being responsible for 90% of the infections, and iii) *G. duodenalis* assemblage B was characterised by an extremely high genetic diversity impairing the correct allocation of isolates to sub-assemblages BIII or BIV, a fact that may be due to elevated coinfection rates or the occurrence of genetic recombination. This study represents the largest and most comprehensive molecular investigation evaluating the genetic diversity of *G. duodenalis* in young children with and without diarrhoea conducted in Mozambique to date. In conjunction with previous risk assessment analyses carried out in the country, the data presented here highlights the need of additional population genetic and epidemiological studies to characterise the epidemiology and transmission dynamics of *G. duodenalis* in Mozambique. Of particular interest would be the investigation of the parasite in environmental water samples and in domestic animal species susceptible of acting as suitable reservoirs of human infections.

## Introduction

Diarrheal diseases remain the second leading cause of mortality in children under 5 years of age worldwide, accounting for 9% of the approximately 5.8 million deaths in 2015 [1,2]. Most of these deaths occurred in sub-Saharan Africa and South Asia, where poor hygiene conditions and insufficient sanitation facilities prevail [3]. The recent update on diarrheal burden from the Global Enteric Multicenter Study (GEMS) [4,5] demonstrated that Rotavirus, *Cryptosporidium* spp., enterotoxigenic *Escherichia coli* producing heat stable toxin (ST_ETEC), and *Shigella* were the leading pathogens associated with moderate-to-severe diarrhoea (MSD) in African and Asian children, in contrast to *Giardia duodenalis* that was inversely associated with MSD [4].

*Giardia duodenalis* (synonyms: *G. intestinalis* and *G. lamblia*) is a globally distributed enteric parasite, and the most commonly detected intestinal protozoan in children [6,7]. The natural history following *G. duodenalis* exposure may lead to either asymptomatic or symptomatic infection ranging from acute diarrhoea to chronic syndrome with weight loss and malabsorption. When present, nausea, vomiting, bloating, and abdominal pain are other clinical manifestations commonly reported. In children in endemic regions, early-life infections by *G. duodenalis* have been associated with stunting (low height for age), wasting (low weight for height), and impaired cognitive function [8–10]. However, asymptomatic infections in African children have been commonly reported in several sub-Saharan countries [11–15]. Progression from infection to disease is a multifactorial process involving both host (age, immune and nutritional status, microbiota composition and abundance) and pathogen (burden, genotype, virulence, immunomodulatory potential) determinants whose interactions and exact contribution to the tilting of the health-disease balance are not completely understood [16,17].

Molecular studies designate *G. duodenalis* as a species complex consisting of eight (A–H) assemblages [18–20]. Assemblages A and B have widely been known to infect humans and other mammals [6,19,21–23], while assemblages C, E and F have strong non-human host specificities and only cause sporadic human infections [24–26]. There is also geographical segregation in *G. duodenalis* assemblages. For instance, assemblage A is more prevalent than assemblage B in the Middle East countries compared to Europe, where the opposite occurs [27]. Extensive genetic variation within assemblages has led to the establishment of sub-assemblages (e.g.: AI, AII, AIII; BIII, BIV). Various *G. duodenalis* loci including the genes coding for the small subunit ribosomal RNA (*ssu* rRNA), the β-giardin (*bg*), the glutamate dehydrogenase (*gdh*), the triosephosphate isomerase (*tpi*), and the elongation factor 1-alpha (*EF-1α*) of the parasite, among others, have been used for genotyping and sub-genotyping purposes [19,27]. However, this typing strategy has some disadvantages such as the intrinsic complexity of molecular methods (nested-PCR followed by either restriction length fragment polymorphisms or DNA sequencing) and the requirement for equipment which is not readily available for most laboratories due to elevated costs among other factors. As assemblage typing is a fundamental tool when studying the molecular epidemiology of *G. duodenalis*, simpler methods (such as assemblage-specific PCR protocols) have been developed and implemented in field surveys [21,22,28,29].

In Africa, the molecular epidemiology of *G. duodenalis* remains largely unknown, despite the fact that different PCR-based methods have been employed to detect and molecularly characterise the parasite in faecal material from human (children and adults, healthy and immunocompromised individuals) and animal (livestock, wildlife) sources and, to a much lesser extent, in environmental (mainly water) samples [9]. In humans, reported prevalence ranged from <1% (in HIV-positive and negative individuals) to >62% (in primary school children) [9]. Genotyping studies from human isolates in Africa targeting the *bg*, *gdh*, and *tpi* loci have

identified *G. duodenalis* assemblage B (sub-assemblages BIII and BIV) as the most common genetic variant of the parasite, followed by assemblage A (sub-assemblages AI and AII) [9]. Human infections by canine-specific (C), ungulate-specific (E), and feline-specific (F) assemblages have been less frequently documented, primarily in children and immunocompromised individuals [9].

During the GEMS study in Mozambique, *G. duodenalis* was found to be more frequent among controls than in cases, which may indicate a potential "protective" effect against diarrhoea, as has also been described in other sub-Saharan countries [30–32]. However, early human challenge studies demonstrated strain-dependent differences in the pathogenicity of *G. duodenalis* infections [33]. Subsequent molecular investigations studies have attempted to correlate the genotype of *G. duodenalis* and the occurrence of diarrhoea with inconclusive or even contradictory results [34,35]. However, most of these surveys were not designed as proper case-control studies, making difficult the obtaining of robust data and valid conclusions. Therefore, assessing the molecular diversity of *G. duodenalis* strains circulating is critical to assess the disease burden and transmission dynamics. Herein we describe the frequency and genetic diversity of *G. duodenalis* detected in symptomatic and asymptomatic children from the Manhiça district enrolled in the context of GEMS between 2007–2012 as a first step to understand the molecular epidemiology of this parasite in Mozambique.

## Methods

### Ethics statement

The GEMS protocol, data collection tools and informed consent were reviewed and approved by the Mozambican National Bioethics Committee for Health (Ref: 11/CNBS/07), the ethics committee of Hospital Clinic of Barcelona (Spain; File 2006/3260) and the Institutional Review Board for Human Subject Research at University of Maryland Baltimore (USA). After informing the objectives and characteristics of the study to each child's caretaker, two copies of a written informed consent were signed by those who agreed to participate. One copy was given to the caretaker and the other was stored in a secure archive at the Centro de Investigação em Saúde de Manhiça (CISM). All patient personal data and information was anonymized or de-identified.

### Study site and study design

In Mozambique, the GEMS was conducted by the CISM in 6 health facilities in the Manhiça district [36], located approximately 80 km north of the capital city Maputo, southern Mozambique. The area covers 2,380 km$^2$ and has a subtropical climate, with two distinct seasons: a warm, rainy season from November to April, and the cool and dry season during the rest of the year [37,38]. Since 1996, CISM has been conducting a continuous Health and Demographic Surveillance System–HDSS (current population followed: 203,132 individuals; 46,851 enumerated and geo-positioned households; children under 5 years: 27,504) with regular updates of all demographic events. During the study period the HDSS was covering approximately 95,000 inhabitants [37].

The rationale, study design and methodology of the GEMS have been previously described elsewhere [39,40]; the study comprised 3 years of acute moderate-to-severe diarrhoea (MSD, GEMS1) and one additional year including less-severe diarrhoea (LSD and MSD, GEMS1A) [4,5], however for the Manhiça site in Mozambique, an additional year between GEMS1 and GEMS1A was included to assess the role of HIV on diarrhoeal aetiology and outcome, thus data and samples were collected uninterruptedly over 5 years. GEMS1 was conducted from December 2007 to October 2011, and GEMS1A from November 2011 to November 2012. The

standardized epidemiological and clinical methods for the case-control study, as well as the full definitions have been previously described elsewhere [36,41]. Briefly, all children aged 0–59 months (stratified in three age groups: 0–11 months, 12–23 months, and 24–59 months), presenting with diarrhoea at six sentinel health facilities and meeting inclusion criteria for the study were invited to participate. Community controls (up to three for MSD cases and one for LSD cases) matched to the index case by age, sex, and neighbourhood, were identified using the HDSS databases and enrolled within 14 days after enrolment of the case, and the stool sample collected and sent to the laboratory at CISM [36].

## Stool collection and *G. duodenalis* screening (GEMS)

The stool sample collection and processing protocols were as described elsewhere [36,42]. Samples were collected in sterile flasks and placed in a cool-box (2–8˚C) for up to 6 hours until transported to the laboratory. Sample aliquots were frozen at –80˚C for further testing. Specifically, for *G. duodenalis* detection, the GIARDIA II immunoassay (TECHLAB, Blacksburg, VA, USA) was used as screening test directly on frozen-thawed stools, following the manufacturer's instructions.

## Molecular study

During the 5 years of GEMS in Manhiça 3,754 stool samples were collected from 1,346 diarrheal cases and 2,408 controls. *Giardia duodenalis* was detected by immunoassay in 27% (1,029/3,754) of the samples. The parasite was significantly less common in diarrheal cases (20%, 271/1,346) than in non-diarrheal controls (32%, 758/2,408) ($P <0.001$). All *Giardia*-positive samples ($n = 757$) available at CISM's biobank were retrieved. Genomic DNAs were extracted and purified using the QIAamp DNA Stool Mini Kit (QIAGEN, Hilden, Germany), and used for downstream molecular testing. Fig 1 shows the distribution of samples among diarrheal cases and controls.

## Assemblage identification by conventional multiplex PCR

Detection and differentiation of *G. duodenalis* assemblages A and B were attempted in all extracted DNAs ($n = 757$) by an assemblage-specific multiplex PCR method based on primers targeting the genes coding for a hypothetical protein (*E1-HP*) and protein 21.1 (*C1-P21*) of the parasite [22]. PCR reactions (25 μL) contained 12.5 μL of 2X QIAGEN Multiplex PCR Master Mix, 5 μL of 5X Q solution, 2.5 μL RNase-free water (QIAGEN), 0.2 μM of each primer (S1 Table), and 3 μL of DNA template. PCR protocols were conducted on MasterCycler Nexus Gradient thermocyclers (Eppendorf AG, Hamburg, Germany) starting with an initial denaturation and enzyme activation step at 95˚C for 15 min, followed by 40 cycles of amplification (denaturation at 94˚C for 30 s, annealing at 56.6˚C for 90 s and elongation at 72˚C for 30 s), and a final extension step at 72˚C for 7 min. Laboratory PCR and sequencing confirmed positive DNA samples of *G. duodenalis* assemblages A and B and nuclease-free water were included in each PCR run as positive and no template controls, respectively.

PCR products were electrophoresed at 100 V for 75 min in 2% agarose gels stained with 0.25 μg/mL ethidium bromide solution and photographed using the Gel Doc EZ Documentation System and ImageLab software version 5.2.1 (Bio-Rad Laboratories Inc., Hercules, CA, USA). DNA aliquots from multiplex PCR-positive samples were shipped to the Spanish National Centre for Microbiology (SNCM) in Majadahonda (Madrid) for further genotyping and sub-genotyping analyses.

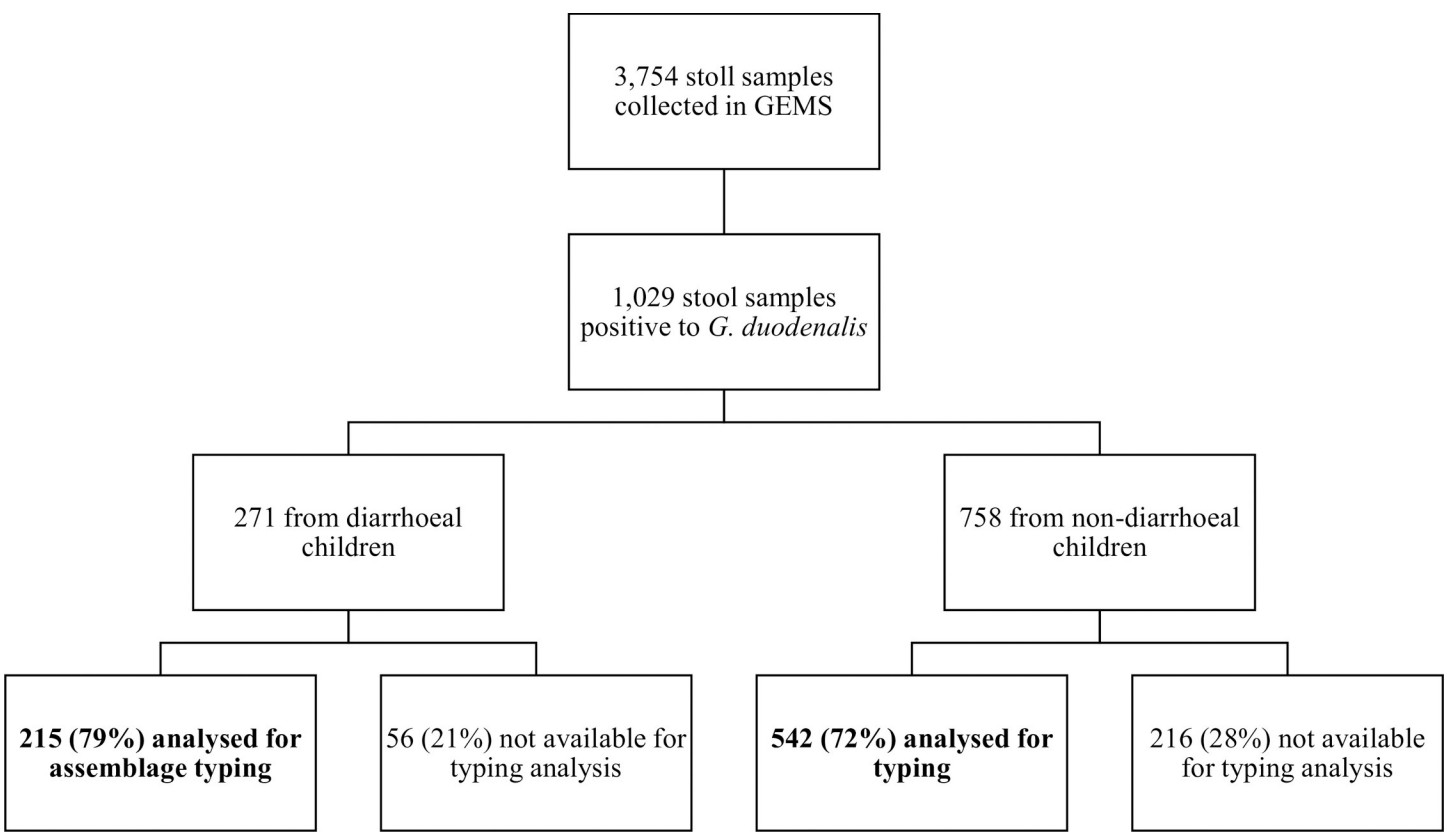

**Fig 1. Study profile of stool samples analysed for molecular typing.**

### Validation of conventional multiplex PCRs

To assess the diagnostic performance of the conventional multiplex PCR assay used here, a subset of aliquoted stool samples (*n* = 171) from the CISM's biobank were randomly selected using matched cases and controls and shipped to the Center for Vaccine Development in Baltimore (USA). These samples were independently processed and analysed using TaqMan Array Card (TAC) qPCR as previously described [43]. The method specifically targets the 18S rRNA (for species identification) and the triosephosphate isomerase (*tpi*, for assemblage differentiation) genes of *G. duodenalis*. All assays were run on customized TACs for the detection of multiple enteric pathogens [43,44].

### Multilocus sequence genotyping (MLSG)

A multi-locus sequence genotyping (MLSG) approach was conducted at the *gdh*, *bg*, and *tpi* genes of the parasite. A semi-nested-PCR protocol was used to amplify a partial fragment (432 bp) of *gdh* using the outer primers GDHe_F and GDHi_R, and the inner primers GDHi_F and GDHi_R (S1 Table) [45]. PCR reaction mixtures (25 μL) consisted of 5 μL of genomic DNA and 0.5 μM of each primer. After an initial denaturation step of 3 min at 95˚C, followed by 35 cycles of amplification (denaturation at 95˚C for 30 s, annealing at 55˚C for 30 s, and elongation at 72˚C for 1 min), and a final extension at 72˚C for 7 min.

A nested-PCR protocol was used to amplify a 511-bp fragment of *bg* using the outer primers G7_F and G759_R and the inner primers G99_F and G609_R (S1 Table) [46]. PCR reactions (25 μL) included 3 μL of genomic DNA and 0.4 μM of each primer. Cycling parameters for the

primary PCR reaction were an initial step of 95˚C for 7 min, followed by 35 cycles of amplification (95˚C for 30 s, 65˚C for 30 s, and 72˚C for 1 min) with a final extension of 72˚C for 7 min. The same conditions were used in the secondary PCR except that the annealing temperature was 55˚C.

A nested-PCR protocol was used to amplify a 530-bp fragment of *tpi* using outer primers AL3543_F and AL3546_R and the inner primers AL3544_F and AL3545_R (S1 Table) [47]. PCR reactions (50 μL) included 2 μL and 2.5 μL of genomic DNA and 0.4 μM of each primer. Cycling parameters for both PCR reactions were an initial step of 95˚C for 5 min, followed by 35 cycles of 94˚C for 45 s, 50˚C for 45 s, and 72˚C for 1 min with a final extension of 72˚C for 10 min.

All PCR protocols described above were conducted on a 2720 thermocycler (Applied Biosystems, CA, USA). Reaction mixes included 2.5 units of MyTAQ DNA polymerase (Bioline GmbH, Luckenwalde, Germany), and 5× MyTaq Reaction Buffer containing 5 mM dNTPs and 15 mM $MgCl_2$. Laboratory PCR-confirmed positive and negative DNA samples for *G. duodenalis* were used as controls and included in each round of PCR. PCR amplicons were visualised on 2% D5 agarose gels (Conda, Madrid, Spain) stained with Pronasafe nucleic acid staining solution (Conda). Amplicons of the expected size were directly sequenced in both directions using the internal primer set described above. DNA sequencing was conducted by capillary electrophoresis using an Applied Biosystems ABI PRISM 3130 automated DNA analyser at the Core Genomic Facility of the Spanish National Centre for Microbiology, Majadahonda (Spain). Sequencing reactions were repeated on samples for which genotyping was unsuccessful in the first instance.

## Statistical analyses

PCR results were entered in a Microsoft Excel spreadsheet and then checked for consistency by independent laboratory personnel. Data on study groups (diarrhoeal *vs.* non-diarrhoeal, MSD *vs.* LSD) and age categories were extracted from the original GEMS dataset. *Giardia duodenalis* assemblages, study groups (diarrhoeal vs. non-diarrhoeal, MSD *vs.* LSD), age categories, and study years were treated as categorical variables and summarized in tables. Differences in frequencies were compared using Chi-squared test or Fisher's exact test as appropriate with significance set at 5%, using Stata version 14 (StataCorp LP, College Station, Texas, USA).

## Sequence and phylogenetic analyses

Raw sequencing data in both forward and reverse directions were visually inspected using the Chromas Lite version 2.1 sequence analysis program (http://chromaslite.software.informer.com/2.1/). Special attention was paid to the detection and recording of ambiguous (double peak) positions. The BLAST tool (http://blast.ncbi.nlm.nih.gov/Blast.cgi) was used to search for identity among sequences deposited in the National Center for Biotechnology Information (NCBI) public repository database. Multiple sequence alignment analyses with appropriate reference sequences were conducted using MEGA 6 to identify *G. duodenalis* assemblages and sub-assemblages, and to annotate the presence of single nucleotide polymorphisms (SNPs) [48]. Detected SNPs were categorized in transition (C↔T and A↔G) mutations, transversion (A↔C, A↔T, G↔C, and G↔T) mutations, and heterozygous (double) peaks at chromatogram inspection.

The evolutionary relationships among the identified *G. duodenalis*-positive samples were inferred by a phylogenetic analysis using the Neighbor-Joining method in MEGA 6. Only sequences with unambiguous (no double peak) positions were used in the analyses. The

evolutionary distances were computed using the Kimura 2-parameter method and modelled with a gamma distribution. The reliability of the phylogenetic analyses at each branch node was estimated by the bootstrap method using 1,000 replications. Representative sequences of different *G. duodenalis* assemblages and sub-assemblages were retrieved from the NCBI database and included in the phylogenetic analysis for reference and comparative purposes. Representative sequences obtained in the present study were deposited in GenBank under the accession numbers MT292375-MT292550 (*gdh* locus), MT332722-MT332817 (*bg* locus) and MT427464-MT427585 (*tpi* locus).

## Results

### Assemblage identification by conventional multiplex PCR

Overall, only 47% (353/757) of the original *Giardia*-positive samples by immunoassay were successfully amplified by conventional multiplex PCR in at least one locus, with 40% (299/757) of them being positive at the *E1-HP* locus and 31% (236/757) at the *C1-P21* locus, respectively; amplifications at both *loci* were achieved in 24% (180/757) of the samples with no result disagreement between them. Assemblage B accounted for 90% (319/353) of all positive samples, followed by assemblage A (8%, 29/353) and mixed A+B infections (1%, 5/353) (Table 1).

### Validation of conventional multiplex PCR by TAC qPCR

To assess the accuracy of assemblage identification, a total of 171 stool samples with a *G. duodenalis* positive-result by immunoassay were independently processed and analysed by conventional multiplex PCR at the *E1–HP* and *C1-P21* loci (Manhiça, Mozambique) and by TAC qPCR at the *tpi* locus (Baltimore, USA). The obtained results by both methods were compared in Table 2. TAC qPCR typed 2-fold more samples than conventional multiplex PCR (83% *vs.* 43%); however, the frequencies of assemblage distribution were similar regardless of the typing strategy followed (B: ~90%; A: ~10%). Mixed A+B infections were detected by TAC qPCR only. Some discrepancies in assemblage assignment were noticed: (i) one A and two B samples by conventional multiplex PCR were identified as B and A (respectively) by TAC qPCR; and (ii) a B sample by conventional multiplex PCR was identified as a mixed A+B infection by TAC qPCR.

### Multi-locus sequence genotyping analyses

Out of the 353 samples with a positive result by conventional multiplex PCR, 351 were available for MLSG analyses. Amplification success rates associated to good-quality sequences were 58% (205/351) for *gdh*, 41% (142/351) for *bg*, and 42% (146/351) for *tpi*, respectively. Overall,

**Table 1. *Giardia duodenalis* assemblage distribution in samples (*n* = 757) analysed by conventional multiplex PCR according to the genetic marker used.**

| Marker | Total–*n* (%)[a] | Assemblage | | |
| --- | --- | --- | --- | --- |
| | | A–*n* (%)[b] | B–*n* (%)[b] | A+B–*n* (%)[b] |
| *E1–HP* | 299 (40) | 22 (7) | 272 (91) | 5 (2) |
| *C1–P21* | 236 (31) | 17 (7) | 217 (92) | 2 (1) |
| Both | 180 (24) | 9 (5) | 169 (94) | 2 (1) |
| At least one | 353 (47) | 29 (8) | 319 (90) | 5 (2) |

a Over 757 samples

b Over the total number of samples amplified at each locus, alone and/or in combination.

**Table 2.** *Giardia duodenalis* assemblage distribution in samples (*n* = 171) analysed by conventional multiplex PCR and TAC qPCR.

| | Conventional multiplex PCR | | | TAC qPCR *n* (%) |
|---|---|---|---|---|
| Assemblage | *E1-HP locus n* (%) | *C1-P21 locus n* (%) | Both loci PCRs *n* (%) | |
| Total[a] | 56 (33) | 53 (31) | 74 (43) | 141 (83) |
| A[b] | 1 (2) | 3 (6) | 4 (5) | 11 (8) |
| B[b] | 55 (98) | 50 (94) | 70 (95) | 127 (90) |
| A+B[b] | 0 (0) | 0 (0) | 0 (0) | 3 (2) |

a Over 171 samples

b Over the total number of samples amplified at each locus or PCR method.

63% (222/351) of samples were genotyped and/or sub-genotyped at least in one of the three markers analysed; MLSG data was obtained from 31% (108/351) of the samples (S2 Table). Sequence analyses revealed the presence of assemblages A (10%; 23/222) and B (90%; 199/222). No mixed infections involving assemblages A+B, or by host-specific assemblages C–H, were detected (Table 3). Within assemblage A, AII was the most prevalent sub-assemblage found (7%, 15/222), followed by ambiguous AII/AIII results (3%, 7/222) and AI (0.5%, 1/222). Within assemblage B, most (59% 132/222) of the sequences corresponded to ambiguous BIII/BIV results. BIII and BIV were present at remarkably similar proportions (13%, 28/222, and 14%, 31/222, respectively). Five samples (2%, 5/222) yielded positive results at the *bg* loci only and their sequences were, therefore, assigned to the assemblage B with unknown sub-assemblage.

Table 3 also shows the *G. duodenalis* assemblage and sub-assemblage distribution, as determined by MLSG at the *gdh*, *bg*, and *tpi* loci, according to the sex and presence/absence of diarrhoea in the children population surveyed.

Regarding sex, assemblage A was present at equal proportions in boys (5%, 12/222) and girls (5%, 11/222), whereas boys (52%, 115/222) were more infected than girls (38%, 84/222) by the assemblage B ($\chi2 = 0.265$, $P = 0.606$). Assemblage distribution was not age-dependent ($P = 0.070$), although assemblage A was rarely seen ($< 1\%$, 1/222) in children younger than 12 months, and most of the assemblage B infections accumulated in children in the age group of 12–23 months (40%, 89/222). Regarding the presence of clinical manifestations (diarrhoea), assemblage A was more prevalent in controls (8%, 18/222) than in cases (2%, 5/222). The same was true for assemblage B (57%, 127/222 *vs.* 32%, 72/222), so no association between the parasite´s genotype and the occurrence of diarrhoea could be demonstrated ($\chi2 = 1.898$, $P = 0.168$).

**Table 3.** *Giardia duodenalis* assemblage and sub-assemblage distribution in samples (*n* = 222) analysed by MLGS at the *gdh*, *bg*, and/or *tpi* loci according to the sex, age group and clinical manifestations of the investigated children population.

| | | | Assemblage A | | | | Assemblage B | | | | |
|---|---|---|---|---|---|---|---|---|---|---|---|
| | | Total | All A | AI | AII | AII/AIII | All B | BIII | BIV | BIII/BIV | Unknown |
| Variable | | *n* (%) | *n* (%) | *n* (%) | *n* (%) | *n* (%) | *n* (%) | *n* (%) | *n* (%) | *n* (%) | *n* (%) |
| Global | | 222 (100) | 23 (10) | 1 (0.4) | 15 (7) | 7 (3) | 199 (90) | 28 (13) | 31 (14) | 135 (61) | 5 (2) |
| Sex | Male | 127 (57) | 12 (5) | 0 (0) | 10 (5) | 2 (0.9) | 115 (52) | 14 (6) | 19 (9) | 78 (35) | 4 (2) |
| | Female | 95 (43) | 11 (5) | 1 (0.4) | 5 (2) | 5 (2) | 84 (38) | 14 (6) | 12 (5) | 57 (26) | 1 (0.5) |
| Age (months) | 0–11 | 44 (20) | 1 (0.5) | 0 (0) | 0 (0) | 1 (0.5) | 43 (19) | 4 (2) | 6 (3) | 33 (15) | 0 (0) |
| | 12–23 | 99 (45) | 10 (5) | 1 (0.5) | 5 (2) | 4 (2) | 89 (40) | 13 (6.0) | 15 (7) | 58 (26) | 3 (1) |
| | 24–59 | 79 (36) | 12 (5) | 0 (0) | 10 (5) | 2 (0.9) | 67 (30) | 11 (5) | 10 (5) | 44 (20) | 2 (1) |
| *Diarrhoea* | Yes (cases) | 77 (35) | 5 (2) | 1 (0.5) | 3 (1) | 1 (0.5) | 72 (32) | 7 (3) | 12 (5) | 52 (23) | 1 (0.5) |
| | No (controls) | 145 (65) | 18 (8) | 0 (0) | 12 (5) | 6 (3) | 127 (57) | 21 (9) | 19 (9) | 83 (37) | 4 (2) |

At the sub-assemblage level, AII was 2-fold more prevalent in males (5%, 10/222) than in females 2% (5/222). This sub-assemblage primarily affected children in the age group of 24–59 months (5%, 10/222) and children without diarrhoea (5%, 12/222). A similar trend was observed for BIII/BIV samples within assemblage B, which were more prevalent in children in the age group of 24–59 months (26%, 58/222) and children without diarrhoea (37%, 83/222).

## Intra-assemblage genetic diversity description

The whole datasets detailing the genetic diversity of the *gdh*, *bg*, and *tpi* representative sequences at the nucleotide level generated in the present study are shown in S3–S5 Tables. Provided information for each sequence included stretch, single nucleotide polymorphisms (SNPs), mutations involved in amino acid change, sample identification number, and Gen-Bank accession number.

At the *gdh* locus 203 out of the 205 available sequences were fully characterised. Out of the 20 assemblage A sequences, one was identified as sub-assemblage AI and showed a SNP in the form of a double peak A/R in position 238 of reference sequence L40509. The other 19 sequences were identified as sub-assemblage AII and were 100% identical to reference sequence L40510 (S3A Table). Virtually all (24/25) BIII sequences were different among them, differing by 1–10 SNPs from reference sequence AF069059. Sites were SNPs tended to accumulate (defined by convenience as hotspots) included positions C99, T147, G150, and C309 of reference sequence AF069059. Some double peak positions and a transversion (C185A) mutation were involved in amino acid change in the protein sequence (S3A Table). Out of the 59 BIV sequences only four had 100% identity with reference sequence L40508, with the remaining 55 differing from it by 1–8 SNPs and most of them representing distinct genotypes of the parasite. Hotspots where SNPs tended to accumulate included positions T183, T387, C396, and C423 of reference sequence L40508. Some double peaks were potentially involved in amino acid change at the polypeptidic chain. No transversion mutations were detected (S3B Table). All 99 sequences with ambiguous BIII/BIV results were different among them, differing also by 4–16 SNPs from reference sequence L40508. Hotspots where SNPs tended to accumulate included positions T135, T183, G186, C255, C273, C345, T366, C372, T387, and A438 of reference sequence L40508. Because of the ambiguity of these sequences, SNPs involved in amino acid change were not determined. No transversion mutations were detected (S3C Table).

At the *bg* locus, a total of 142 sequences were fully characterised. Out of the 14 assemblage A sequences, six belonged to AII and eight to AIII. Five of the AII sequences were identical to reference sequence AY072723, with the remaining one differing from it by two SNPs. Seven of the AIII sequences were identical to reference sequence AY072724, with the remaining one differing from it by two transversion (C426G and C525A) mutations. A large genetic variability was observed within the 128 sequences assigned to B (S4 Table); of them, eight sequences had 100% identity with reference sequence AY072727. The remaining 120 B sequences differed from AY072727 by 1–8 SNPs, including a sequence with a C insertion in position 339 that introduced a stop codon. Nine sequences showed a distinctive SNP pattern A183, C309, T519, and C564. These positions, together with C165 were the main hotspots in B. Most of the remaining sequences represented variations in the number or presentation (mutation, double peak) of these SNPs. Several transition mutations and ambiguous positions were involved in amino acid change. No transversion mutations were detected within B sequences (S4 Table).

At the *tpi* locus a total of 146 sequences were fully characterised. All 11 assemblage A sequences were identified as AII. Of them, 10 were identical to reference sequence U57897, and the remaining one differed from it by a single transversion (C394R) mutation. Out of the

85 sequences assigned to BIII, eight showed 100% identity with reference sequence AF069561. The remaining 77 sequences had a large genetic diversity and were distributed in 72 distinct genotypes, most of them presenting different combinations of the hotspot positions C34, G105, C108, C141, and G153. Several double peaks, transition, and transversion mutations were involved in amino acid change at the protein level (S5A Table). Similarly, all 12 sequences identified as BIV were different among them, differing by 2–11 SNPs with reference sequence AF069560. Hotspot positions included A176 and A395. Several double peaks, transition, and transversion (A437C) mutations were associated with amino acid change in the protein chain (S5B Table). In line with the very high degree of genetic heterogeneity observed within BIII and BIV sequences, virtually all (35/38) sequences with a BIII/BIV ambiguous result were different among them, differing from reference sequence AF069560 by 4–15 SNPs. Consequently, a large proportion of these SNPs (mostly double peaks) were potentially involved in amino acid change at the protein level (S5C Table).

### Intra-assemblage B genetic diversity analysis

Independently of the molecular marker used, genetic diversity was far higher within assemblage B than within assemblage A sequences. Multiple sequence alignments of BIII, BIV, and ambiguous BIII/BIV sequences at the *gdh*, *bg*, and *tpi* loci revealed the presence of SNPs in multiple sites across used reference sequences, varying from 28 (for BIV sequences at the *tpi* locus) to 88 (for B sequences at the *bg* locus) sites (S6 Table). Overall, 2,225 SNPs were identified among assemblage B sequences in all three loci. Of them, 37% (821/2,225) corresponded to transition and (to a much lower extend) transversion mutations and nucleotide insertions, and 63% (1,404/2,225) to double peaks. Identified hotspot sites (see section above) at each *locus* accumulated the bulk of these SNPs (63%; 1,392/2,225).

The distribution of transition/transversion mutations and double peaks differed substantially among sub-assemblages and loci (Fig 2). At the *gdh* locus, hotspot sites accumulated 50–55% of all SNPs detected in BIII and BIV sequences, but this figure increased to 74% in BIII/BIV sequences. Double peaks accounted for 26–38% of the SNPs detected in BIII and BIV sequences, but for 82% of the ambiguous BIII/BIV sequences (Fig 2A). At the *tpi* locus, hotspot sites accumulated 33–41% of all SNPs detected in BIII and BIV sequences, but this figure increased to 60% in BIII/BIV sequences. Remarkably, double peaks accounted for 60–68% of all SNPs detected in BIII and ambiguous BIII/BIV sequences, but only for 5% in BIV sequences (Fig 2B). Finally, at the *bg* locus, hotspot sites concentrated 60% of the SNPs detected in assemblage B, sequences, of which 54% corresponded to double peaks (Fig 3A).

### Phylogenetic analyses

The Fig 3 shows the evolutionary relationships among the *G. duodenalis* sequences generated in the present study at the *gdh* locus. For comparative purposes, similar sequences generated by our research team in areas of high (Ethiopia, Angola, and Brazil) and low (Spain) endemicity were also included in the analysis. Assemblage A sequences grouped together in well-defined clusters with appropriate reference sequences. Assemblage B sequences also formed an independent branch of the phylogenetic tree, but sub-assemblage BIII and BIV sequences could not be segregated in independent clusters. Phylogenetic trees generated at the *bg* (S1 Fig) and *tpi* (S2 Fig) loci corroborated this finding.

## Discussion

This study represents the largest and most comprehensive molecular investigation evaluating the genetic diversity of *G. duodenalis* in young children with and without diarrhoea conducted

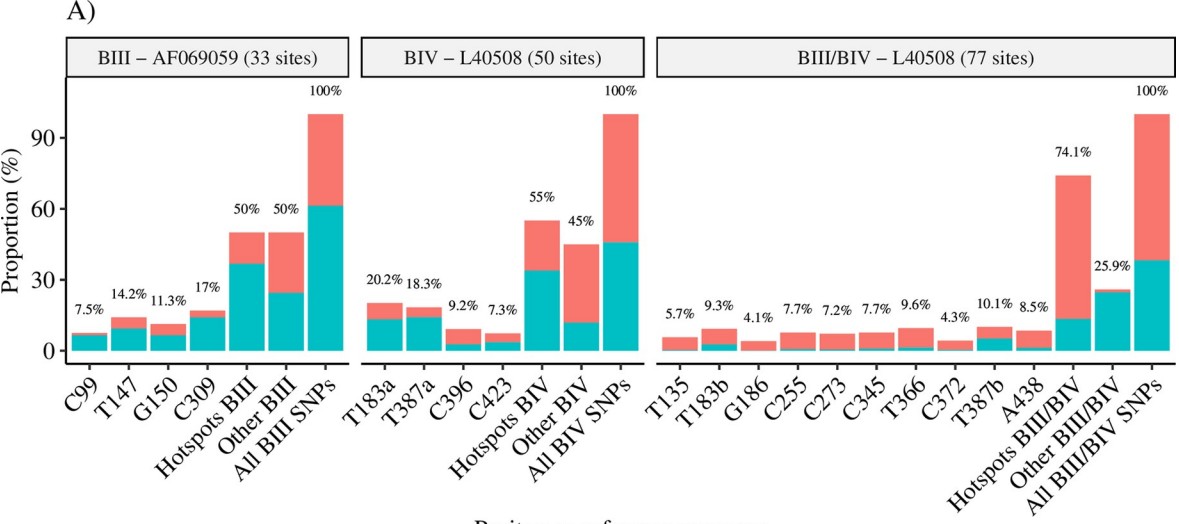

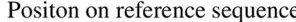

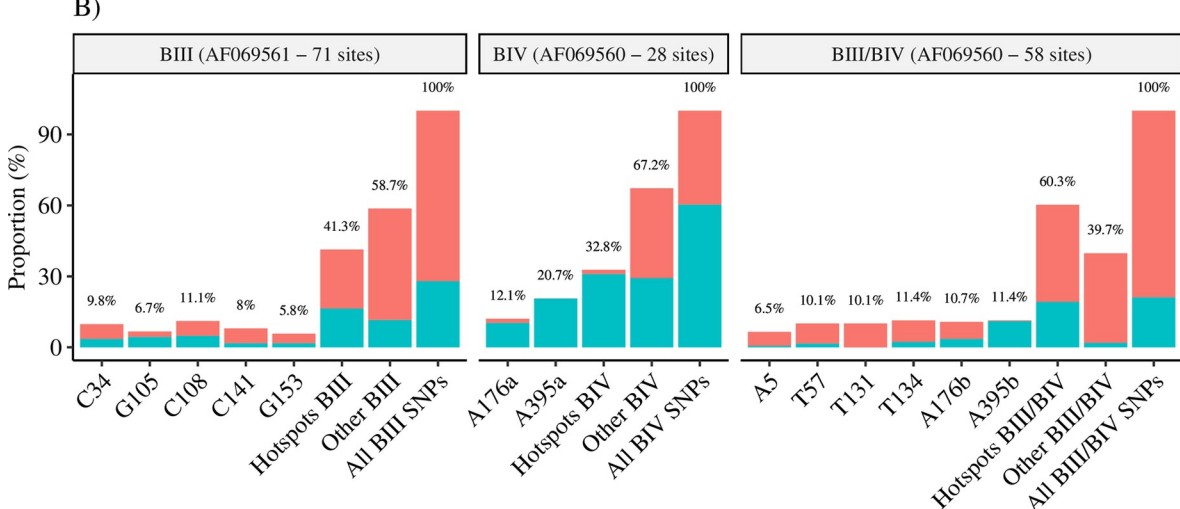

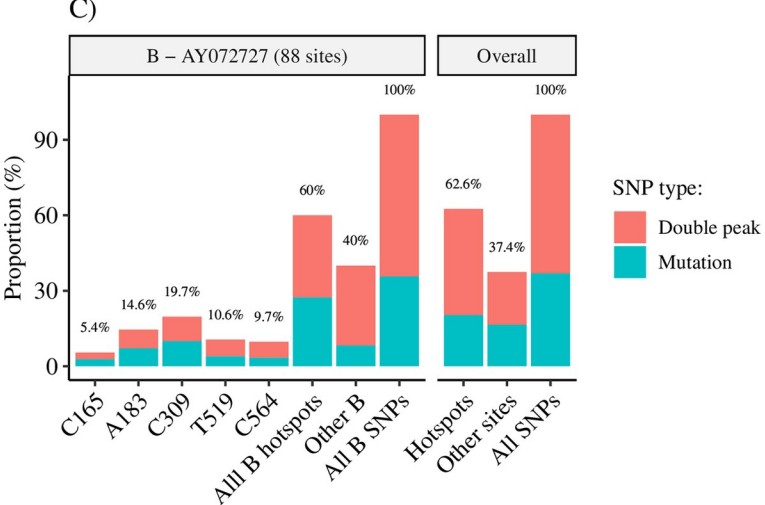

**Fig 2. Distribution of single nucleotide polymorphisms segregated by mutations and double peaks, in *Giardia duodenalis* assemblage B sequences according to the genetic marker considered.** (A) SNPs at the glutamate dehydrogenase (*gdh*) locus; (B) SNPs at the triose phosphate isomerase (*tpi*) locus; (C) SNPs at the ß-giardin (*bg*) locus and overall figures for all assemblage B sequences.

in Mozambique to date. The survey provides three relevant contributions to the understanding of the epidemiology of human giardiasis in the country. First, early childhood diarrhoea by *G. duodenalis* was not linked to a given assemblage of the parasite. Second, assemblage B caused nine out of ten infections by *G. duodenalis* in young children. And third, an extremely high molecular diversity at the nucleotide level was observed within assemblage B, but not within assemblage A sequences, independently of the genetic marker investigated.

*Giardia duodenalis* infections do not seem to be positively associated with acute diarrhoea in young children living in endemic areas in low- and medium-income countries [49,50], or have been even described as "protective" against it [51,52]. Indeed, molecular epidemiological case-control studies carried out in Ivory Coast [32], Central African Republic [30], Tanzania [31], in addition to the GEMS project [4], have demonstrated that *G. duodenalis* was significantly more prevalent in controls than in cases. This is the reason why the parasite was systematically absent in global burden estimations of diarrhoeal disease [53]. The genotype of *G. duodenalis* involved in these infections may play a major role in the triggering (or not) of clinical manifestations associated to gastrointestinal illness.

Several molecular-based surveys have attempted to correlate the presence of diarrhoea with the genotype of *G. duodenalis*. In a seminal study carried out in patients (*n* = 18) between 8 and 60 years of age in The Netherlands, a strong correlation was found between assemblage A and intermittent diarrhoea, and assemblage B and persistent diarrhoea. Based on statistical analyses of epidemiological data, the authors inferred that assemblage A isolates would be more prevalent in asymptomatic infected individuals [35]. The opposite result was observed in a subsequent study conducted in children (*n* = 23) under 5 years of age attending day-care centres in Australia, where assemblage B genotypes were more frequently found in asymptomatic children than those of assemblage A [54]. Close associations between assemblage A and diarrhoea have also been demonstrated in clinical patients (*n* = 44) in Turkey [55], and in outpatient children (*n* = 43) less than 5 years of age with and without clinical manifestations in Spain [34]. This very same result was also confirmed in a large prospective case-control study targeting individuals (*n* = 343) of all ages with giardiasis in Bangladesh [56]. Inconclusive data or no correlation between genotypes and diarrhoea were documented in other surveys carried out in Brazil, India, and Iran [57–59]. Except for the study by Haque *et al*s. [56], most of these investigations lacked a proper case-control design and genotyped a relatively low (typically less than 50) number of *G. duodenalis* isolates belonging to different age groups and clinical backgrounds, making difficult the obtaining of robust, unquestionable data. In this context, the present study took advantage of a large (*n* = 222) panel of genotyped cases and controls stratified by age and sex to demonstrate that diarrhoea was not associated to a given *G. duodenalis* assemblage in Mozambican children under five years of age. The survey represents the most thorough attempt to investigate the potential link between the parasite´s genotype and diarrhoea in the African continent. This finding arises the question of whether the distribution of *G. duodenalis* assemblages in infected humans may follow an age-related pattern, and, if this is the case, whether age-driven differences in assemblage predominance may account for a higher likelihood of diarrhoeal illness. Indeed, children have been demonstrated to be more commonly infected by assemblage B (83%, 44/53) than adults (52%, 22/42) in clinical patients between 1 and 75 years of age in Spain [60]. In this very same country, assemblage B was significantly more prevalent than assemblage A in asymptomatic outpatient children, but this

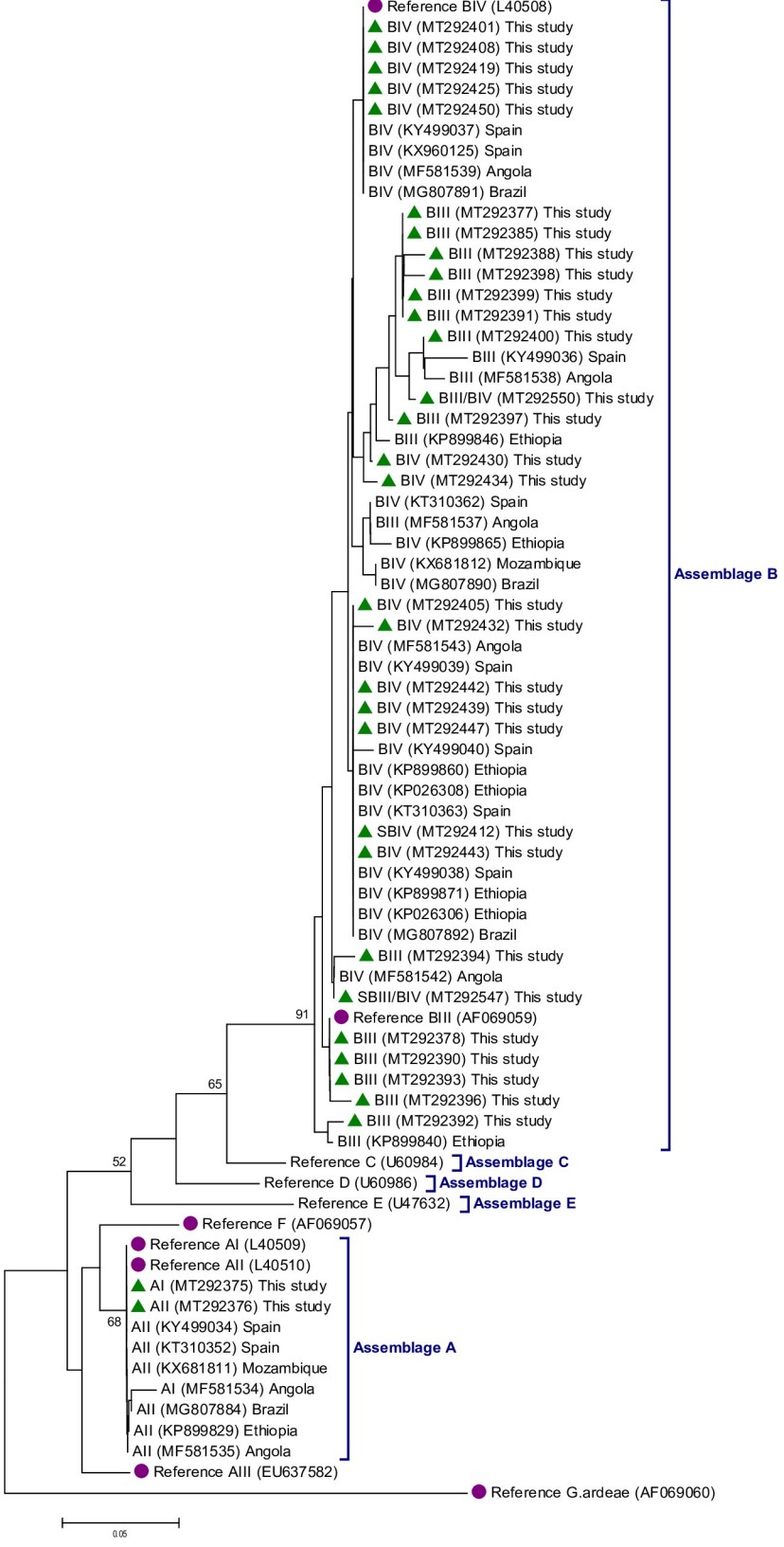

**Fig 3. Phylogenetic tree depicting evolutionary relationships among *Giardia duodenalis* sequences at the *gdh locus* from Mozambican children.** The analysis was inferred using the Neighbor-Joining method of the nucleotide

sequence covering a 416-bp region (positions 76–491 of GenBank: L40508) of the gene. Bootstrap values lower than 50% are not shown. Cyan filled circles represent reference sequences downloaded from the GenBank database; filled dark green triangles represent sequences generated in the present study. *Giardia ardeae* was used as the outgroup.

effect disappeared when individuals of older ages were included in the analysis [34]. Definitively, more research should be conducted in adult populations in Mozambique and other African countries to clarify the extent to which age may be a relevant factor in the distribution and virulence of *G. duodenalis* assemblages in human infections.

Molecular data presented here confirmed that assemblage B was far more prevalent than assemblage A (90% *vs*. 10%) in children younger than five years in Mozambique, regardless of the presence/absence of clinical manifestations. In the only genotyping study available in the country so far, sub-assemblages AII and BIV were detected in two HIV-positive patients in the Gaza province [61]. The above-mentioned result is in line with those previously reported in most African countries demonstrating the predominance of assemblage B over assemblage A [9]. Indeed, assemblage B has been identified at frequencies > 80% in Egypt [62], Ethiopia [63], Guinea-Bissau [64], Kenya [65], Morocco [66], Ruanda [67], and Tanzania [68]. The opposite has been proven true in few surveys showing that assemblage A was more prevalent than assemblage B in Ethiopia [25] and the Democratic Republic of São Tomé and Príncipe [69]. The discrepancies observed in the geographical segregation of *G. duodenalis* genotypes may be indicative of different sources of infection and transmission pathways. In this regard, canine- (C, D), ungulate- (E), or feline-specific (F) assemblages were not detected in the present study. This apparent lack of animal-specific genetic variants of the parasite should be interpreted with caution, as our initial typing strategy was designed to prioritize the detection of assemblages A and B. It is possible that isolates belonging to C-F assemblages were undetected under our typing scheme. It should be noted that assemblage C has been described in a cancer patient in Egypt [62], assemblage F in symptomatic and asymptomatic individuals in Ethiopia [25], and assemblage E in three independent studies in Egypt [70–72].

Confirming the results obtained in previous molecular epidemiological studies conducted in other African countries such as Angola [73], Egypt [62], Ethiopia [25,63,74], Kenya [65], and Uganda [11], this study highlights the extraordinary genetic variability within *G. duodenalis* assemblage B also in Mozambique. This finding was particularly evident at the *gdh* and *tpi* markers, for which the vast majority (85–100%) of the generated BIII, BIV, and BIII/BIV sequences differed among them at the nucleotide level. Interestingly, some of the SNPs detected were associated with single amino acid substitution in the coding regions of the investigated gene, potentially leading to changes in the structure or function of the protein. When compared to selected reference sequences, obtained sequences that were unmistakably assigned to BIII and BIV tended to vary only in 2 to 5 positions (hotspots) either as mutations or ambiguous (double peak) sites. In these sets of hotspots, the proportion of sites involving double peaks varied greatly from 26–38% at the *gdh* locus, and from 5–60% at the *tpi* locus. Interestingly, these percentages increased in all cases over 68% in ambiguous BIII/BIV sequences, explaining why these isolates were difficult to allocate to a given sub-genotype. Two independent mechanisms have been proposed to explain the presence of double peaks. The first one would imply the occurrence of true mixed infections involving different genetic variants (both at the assemblage and the sub-assemblage levels) of *G. duodenalis*. An elevated proportion of coinfections would be expected in highly endemic areas for the parasite such as the one under investigation in the present study. The second one would imply the occurrence of genetic recombination associated to low levels of allelic sequence heterozygosity (ASH) [75,76]. As in the case of other binucleate diplomonads, *G. duodenalis* has been largely

considered as a strict asexual (clonal) organism. Under this premise, allelic copies both within and between nuclei would be expected to diverge over time as a consequence of the accumulation of mutations, resulting in increased ASH values [77]. This is not the case of *G. duodenalis* assemblage A isolates, characterised by low (< 0.01%) ASH values [75,78]. These findings were confirmed in sub-assemblages AI and AII, and, to a much lesser extent in assemblage B isolates [79]. Additional evidence of genetic recombination events has been demonstrated within assemblage B in single (trophozoite and cyst) cells [80] and within sub-assemblages BIII and BIV at the genetic population level [81]. All together, these data clearly indicate the presence of a genetic homogenization mechanism in *G. duodenalis*. Those proposed include intragenic recombination, intra-allelic sequence exchange, or diplomixis during the cystic stage [77,82,83]. The exact contribution of each of these potential mechanisms to the genetic diversity of *G. duodenalis* remains unknown. In the case of suspected mixed infections, cloning of PCR amplicons or next generation sequencing methods may help in detecting genetic variants of the parasite that are underrepresented in the population pool, and that are otherwise undetectable using conventional PCR methods and Sanger sequencing [84–86]. These approaches have not been adopted here, but it would be interesting to do it in selected samples in the near future.

Among the strengths of this study are the carefully designed recruitment of case-control pairs stratified by age and sex, the large number of isolates successfully genotyped, and the adoption of a MLSG scheme to accurately determine the genetic diversity of *G. duodenalis* at the genotype level. However, some of the results obtained and the conclusions reached in it may be biased by methodological limitations. First, we did not attempt to provide a full epidemiological description including association analyses between *G. duodenalis* genotypes and sociodemographic and clinical characteristics of the participating children, a task to be conducted in an independent study. Second, barely one in two *Giardia*-positive samples by the immunoassay used as initial screening diagnostic method were subsequently confirmed by conventional multiplex PCR. The limited correlation between both methods may be indicative of a higher than expected rate of false-positive results by immunoassay, or of amplification failure due to insufficient primer specificity as a consequence of the extremely high genetic diversity within *G. duodenalis* isolates revealed at the *gdh*, *bg*, and *tpi* loci. Suboptimal removal of PCR inhibitors (proteases, DNases, polysaccharides, bile salts) may also account, to a variable degree, for the latter possibility. Differences in primer design and PCR performance may also explain the discrepant rates of mixed A+B infections obtained by TAC qPCR and the typing (*gdh*, *bg*, *tpi*) PCRs used in this study. Third, the human-oriented design of the conventional multiplex PCR protocol used as confirmatory test did not allow the detection of animal-specific assemblages (C-F) of the parasite, so the contribution of zoonotic transmission and environmental reservoirs to the burden of human giardiasis remains unknown. Fourth, ambiguous (particularly BIII/BIV) sequences potentially associated to true mixed infections can be further investigated either by cloning and sequencing of PCR products or by next generation sequencing analyses. Because none of these methodologies were available in our laboratory, this remaining task should be conducted in future molecular studies.

Overall, the high level of genetic diversity observed within *G. duodenalis* isolates in Mozambican children is compatible with an hyperendemic epidemiological scenario characterised by elevated rates of infection and re-infection, and very likely, high levels of environmental contamination. In this regard, a recent risk association study conducted in the province of Zambézia revealed that drinking untreated water and having regular contact with domestic animals were major risks for acquiring protist infections including giardiasis [15]. Supporting these findings, a recent quantitative microbial risk assessment analysis has estimated that consumption of unsafe water causes 2 million of cryptosporidiosis cases and $1.6 \times 10^5$ disability-

adjusted life years in Mozambique annually [87]. These results were in line with those previously obtained in the country highlighting the urgent need of improving access to safe drinking water and sanitary conditions to minimize the risk of environmental contamination and the transmission of pathogens transmitted by the faecal-oral route [41,88,89].

## Conclusion

In this study no obvious differences in the distribution of *G. duodenalis* assemblages were observed in infected children younger than five years of age with and without diarrhoea in Mozambique, demonstrating that the occurrence of gastrointestinal illness was not associated to a given genotype of the parasite. Assemblage B was the most prevalent genetic variant circulating in the surveyed paediatric population, causing nine out of ten of the genotyped cases. An extremely high genetic diversity was observed within assemblage B isolates, impairing the correct identification of sub-assemblages BIII and BIV. This diversity was associated, to a great extent, to the presence of ambiguous positions (double peaks) at the chromatogram level, suggesting that coinfections and genetic recombination events were simultaneously taking place in the investigated population. The exact contribution of these mechanisms to the genetic variability of *G. duodenalis* are currently unknown. Based on previous risk assessment studies conducted in the country, waterborne and zoonotic transmission are thought to be common routes of human infections. Since no data are currently available, there is an urgent need to conduct molecular studies based on PCR and Sanger sequencing to identify and genotype diarrhoea-causing protist species including *G. duodenalis* in environmental water samples and in domestic animal species. This information is essential to elucidate sources of infection and transmission pathways of these pathogens.

## Supporting information

**S1 Table. List and sequence of primers used for molecular identification of *G. duodenalis* assemblages and multilocus sequence genotyping.**
(DOCX)

**S2 Table. Multilocus sequence genotypes of *G. duodenalis* in the children population from Manhiça, Mozambique (2007–2012).** Provided information for each sample includes unique identifiers, age group, study year and study groups.
(XLSX)

**S3 Table. Diversity, frequency, and molecular features of *G. duodenalis* sequences at the glutamate dehydrogenase *locus* obtained in the children population from Manhiça, Mozambique (2007–2012).** Single Nucleotide Polymorphisms (SNPs) inducing amino acid substitutions are highlighted in red indicating the amino acid change. Nucleotide position / sites at reference sequences with SNPs are also presented, with hotspots for SNPs shaded in yellow. GenBank accession numbers are provided.
(XLSX)

**S4 Table. Diversity, frequency, and molecular features of *G. duodenalis* sequences at the β-giardin *locus* obtained in the children population from Manhiça, Mozambique (2007–2012).** Single Nucleotide Polymorphisms (SNPs) inducing amino acid substitutions are highlighted in red indicating the amino acid change. Nucleotide position / sites at reference sequences with SNPs are also presented, with hotspots for SNPs shaded in yellow. GenBank accession numbers are provided.
(XLSX)

**S5 Table. Diversity, frequency and molecular features of *G. duodenalis* sequences at the triose phosphate isomerase *locus* obtained in the children population from Manhiça, Mozambique (2007–2012).** Single Nucleotide Polymorphisms (SNPs) inducing amino acid substitutions are highlighted in red indicating the amino acid change. Nucleotide position / sites at reference sequences with SNPs are also presented, with hotspots for SNPs shaded in yellow. GenBank accession numbers are provided.
(XLSX)

**S6 Table. Intra-assemblage B single nucleotide polymorphisms distribution and classification among *G. duodenalis* sequences at the *gdh*, *bg* and *tpi loci* from Mozambican children.** Hotspots for SNPs are identified and the summarised ***comparison*** of frequencies between hotspot and non-hotspot sites are highlighted with darker shades.
(XLSX)

**S1 Fig. Phylogenetic tree depicting evolutionary relationships among *G. duodenalis* sequences at the *bg locus* from Mozambican children.** The analysis was inferred using the Neighbor-Joining method of the nucleotide sequence covering a 517-bp region (positions 96–612 of GenBank: AY072727) of the gene. Bootstrap values lower than 50% are not shown. Cyan filled circles represent reference sequences downloaded from the GenBank database; filled dark green triangles represent sequences generated in the present study. No outgroup sequence was used as *bg* is a *Giardia*-specific gene.
(TIF)

**S2 Fig. Phylogenetic tree depicting evolutionary relationships among *G. duodenalis* sequences at the *tpi locus* from Mozambican children.** The analysis was inferred using the Neighbor-Joining method of the nucleotide sequence covering a 479-bp region (positions 1–479 of GenBank: AY069560) of the gene. Bootstrap values lower than 50% are not shown. Cyan filled circles represent reference sequences downloaded from the GenBank database; filled dark green triangles represent sequences generated in the present study. *Giardia muris* was used as the outgroup.
(TIF)

## Acknowledgments

We thank the children and their caretakers who participated in the study, as well as the clinical, field and laboratory staff who worked tirelessly to ensure the data collection and laboratory testing was performed according to the standardized protocol. We also thank all the local government authorities (district Administration and Health Directorate) and all community leaders for supporting and collaborating in the study.

## Author Contributions

**Conceptualization:** David Carmena, Inácio Mandomando.

**Data curation:** Augusto Messa, Jr., Pamela C. Köster, Marcelino Garrine, David Carmena, Inácio Mandomando.

**Formal analysis:** Augusto Messa, Jr., Pamela C. Köster, David Carmena, Inácio Mandomando.

**Funding acquisition:** Karen Kotloff, Myron M. Levine, Pedro L. Alonso, David Carmena, Inácio Mandomando.

**Investigation:** Augusto Messa, Jr., Pamela C. Köster, Marcelino Garrine, Tacilta Nhampossa, Sérgio Massora.

**Methodology:** Carol Gilchrist, Luther A. Bartelt, Karen Kotloff, Myron M. Levine, Pedro L. Alonso, David Carmena, Inácio Mandomando.

**Project administration:** David Carmena, Inácio Mandomando.

**Resources:** Karen Kotloff, Myron M. Levine, Pedro L. Alonso, David Carmena, Inácio Mandomando.

**Software:** Augusto Messa, Jr., Pamela C. Köster.

**Supervision:** Marcelino Garrine, Tacilta Nhampossa, Sérgio Massora, David Carmena, Inácio Mandomando.

**Validation:** Marcelino Garrine, Carol Gilchrist, Luther A. Bartelt, David Carmena, Inácio Mandomando.

**Visualization:** Augusto Messa, Jr., Carol Gilchrist, Luther A. Bartelt, Karen Kotloff, Myron M. Levine, Pedro L. Alonso, David Carmena, Inácio Mandomando.

**Writing – original draft:** Augusto Messa, Jr., David Carmena, Inácio Mandomando.

**Writing – review & editing:** Augusto Messa, Jr., Pamela C. Köster, Marcelino Garrine, Carol Gilchrist, Luther A. Bartelt, Tacilta Nhampossa, Sérgio Massora, Karen Kotloff, Myron M. Levine, Pedro L. Alonso, David Carmena, Inácio Mandomando.

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
