## [Decision Letter · Decision Letter 0]

31 Oct 2020

Dear PhD Mandomando,

Thank you very much for submitting your manuscript "Molecular diversity of Giardia duodenalis in children under 5 years from the Manhiça district, Southern Mozambique enrolled in a matched case-control study on the etiology of diarrhea" for consideration at PLOS Neglected Tropical Diseases. As with all papers reviewed by the journal, your manuscript was reviewed by members of the editorial board and by several independent reviewers. The reviewers appreciated the attention to an important topic. Based on the reviews, we are likely to accept this manuscript for publication, providing that you modify the manuscript according to the review recommendations. 

Sincerely,

Rojelio Mejia, M.D.

Associate Editor

Steven Singer

Deputy Editor

Reviewer's Responses to Questions

**Key Review Criteria Required for Acceptance?**

**Methods**

-Are the objectives of the study clearly articulated with a clear testable hypothesis stated?

-Is the study design appropriate to address the stated objectives?

-Is the population clearly described and appropriate for the hypothesis being tested?

-Is the sample size sufficient to ensure adequate power to address the hypothesis being tested?

-Were correct statistical analysis used to support conclusions?

-Are there concerns about ethical or regulatory requirements being met?

Reviewer #1: In this study, it is proposed to establish the correlation between the symptoms (diarrhea) and the genotype of the Giardia causing the infection. The results indicate that the relationship between the two is neither obvious nor significant, however they find a great genetic diversity among isolates of genotype B, based on previous works (Infection, Genetics and Evolution Volume 47, January 2017, Pages 155-160).

1. They mention that mainly in children this one finds genotype B, while in adults genotype A is found more frequently, however the studies carried out in this research only include samples from children.

2. They must specify from where they take the characteristics to classify the samples in the BIII / BIV subgenotypes.

3. According to the methodology they carry out, is it possible that the ambiguities in the classification are due to the presence of other genotypes (C-H) that are not searched with this procedure?

4. The presence of double peaks in the electropherogram can be removed with further purification of the genetic material? This would reduce the ambiguities in the classification

1. The research aims to study the relationship between digestive pathology (diarrhea) in children and the presence of the duodenal Giardia genotype in their stools

1.2 . According to the information presented, they do not find statistically significant differences between genotype and diarrhea in children. The results show that they only found type B. It is striking that group A only appears in adults. According to the title of the work, it was planned to work only children because they included adults, would it be convenient to explain what was the reason for doing it. The exposed bibliography is updated. However, it does not speak of genetic diversity where it finds the BIII / BIV groups with results of double peaks that are interpreted as ambiguities and that allows them to think that they are different genotypes. From what I consider, it must present a difference in the electrophoretic shift electropherogram that supports this work as it is not well documented solidly. We must remember the genomic plasticity of Giardia makes it an interesting protozoan

1.3. They do not mention the hypothesis of the work that should be included if they seek to relate the presence of diarrhea and the Giardia genotype if they want to postulate the genotypes with the presence of diarrhea in children

 1.4.The size of the sample is sufficient. However, they do not explain how it is calculated, one of the parameters that must be considered is the prevalence of Giardiasis in their country

1.5.They include statistical treatment where it is shown that no relationship between the parameters studied

1.6.There are no ethical problems at work

Reviewer #2: (No Response)

**Results**

-Does the analysis presented match the analysis plan?

-Are the results clearly and completely presented?

-Are the figures (Tables, Images) of sufficient quality for clarity?

Reviewer #1: The use of an enzymatic method to demonstrate the presence of the parasite is not conclusive because it is showing two different parametrons in theenzymatic method te presence of antibodies that is not always conclusive because it depends on the antigenic of the parasite's proteins. You must put the coproparasitoscopic in where the presence of the parasite is demonstrated. This is probably due to the different positivity when the gdhtpibg gene initiators were used, the results are shown in Table 3.

Reviewer #2: (No Response)

**Conclusions**

-Are the conclusions supported by the data presented?

-Are the limitations of analysis clearly described?

-Do the authors discuss how these data can be helpful to advance our understanding of the topic under study?

-Is public health relevance addressed?

Reviewer #1: I am not going to refer to your results obtained when you include adults because that is not the purpose of the study. I include the limitations and discussion of your study. Of course, giardiasis is a parasite of public health importance. But trying to give an explanation for the presence of diarrhea with the genotype without any other clinical data is not possible, it is also fully demonstrated that there is no relationship for the plasticity of the Giardia genome. Making conclusions about transversion and mutations is very risky, you have to be very careful about any change in SNPs that involve an amino acid and the change of a single nucleotide is not enough. Their results showed that in 8 sequences there was no change. It is difficult to follow it because it finally concludes that group B did not present transversion or mutations within group B (TABLE S 4

They conclude that the study represents an investigation trying to evaluate the genetic diversity of Giardia in children with and without diarrhea first point the genotype of the parasite was not correlated with diarrhea in children Which is demonstrated by finding in 9 out of 10 children agrees that it is not statistically significant The third point, the explanation of the diversity found in the Giardia III / Giardia IV genotypes must present the cleanest technique in its results without ambiguities, no double peaks and the purity of the DNA used

Reviewer #2: (No Response)

**Editorial and Data Presentation Modifications?**

Reviewer #1: I have no comments

Reviewer #2: (No Response)

**Summary and General Comments**

Reviewer #1: The rationale for classifying Giardia into genotypes should be specified, because changes in electropherograms with a single nucleotide are not enough. The presence of ambiguities and double peaks that do not give solidity to these results, generally this occurs when lack of purity in the DNA to demonstrate the purity of the DNA

Reviewer #2: This is a very important paper for the Giardia field, that in addition to bring insights on the lack of relationship between assemblage and clinical disease, performs a large-scale genotyping in samples from Mozambique, constituting an important source of epidemiological information for that country. After careful revision of the manuscript PNTD-D-20-01460 I have only some minor comments that might be useful to improve the current version.

• Table 2 demonstrates how the TAC qPCR was superior in sensitivity than the PCR performed in Mozambique. Could the authors please add in the Discussion whether the publication of this articles can impact – or not – changes in the national policy – if any-, for epidemiological characterization/reporting of Giardia infections.

• Do the authors have an explanation to why mixed infections were determined only by TAC qPCR but not with genotyping? If this is important, please add this response to the Discussion.

• I would remove lines 506 to 512 - we do not want to speculate findings for other assemblages if the assay was designed to focus more on detection of assemblages A and B.

• Abstract: at line 28, please clarify to the wide readership of PNTD what are “cases” in the GEMS study.

• There is a typo at line 67

• Lines 109, 110: the following sentence is not necessary in my opinion: “which are referred by using Roman numerals as suffixes”.

• Lines 163 to 167 -Please clarify or give more detail to the reader: if the author said that there were 3 years of GEMS1 + 1 year of GEMS1A, please explain how this study consisted in 5 years. 

• Line 77, please remove “as”.

• Please replace, after the first appearance, Giardia duodenalis for G. duodenalis, to make the reading sound smoother. Also, it will standardize the text.

• In Figure 1, “positive” is misspelled in one of the boxes

• Line 195, replace “was” for “were”

• Line 315 – please define the abbreviation here in the title

• Sentence at line 362 “Hotspots were…” can be rephrased for clarity.

• Please choose “CHARACTERIZED” or CHARACTERISED”

• Line 427 – Typo in “giardian”

• Line 500, “The above mentioned result is in line with those previously reported in most African countries” – I would remove this sentence or explain better what the authors are trying to say here.

• Line 553 – missing “d” in sociodemographic

This is an important article, and if those small comments are addressed, I recommend the paper for publication.

Camila Coelho, PhD

NIAID/NIH

PLOS authors have the option to publish the peer review history of their article (what does this mean?). If published, this will include your full peer review and any attached files.

Reviewer #1: No

Reviewer #2: Yes: Camila H. Coelho
---

## [Editor Report · Decision Letter 1]

18 Nov 2020

Dear PhD Mandomando,

We are pleased to inform you that your manuscript 'Molecular diversity of Giardia duodenalis in children under 5 years from the Manhiça district, Southern Mozambique enrolled in a matched case-control study on the etiology of diarrhea' has been provisionally accepted for publication in PLOS Neglected Tropical Diseases.

Best regards,

Rojelio Mejia, M.D.

Associate Editor

Steven Singer

Deputy Editor

Please address line 475 "Spaon"? not sure what this means.

---

## [Editor Report · Acceptance letter]

12 Jan 2021

Dear PhD Mandomando,

We are delighted to inform you that your manuscript, "Molecular diversity of * Giardia duodenalis * in children under 5 years from the Manhiça district, Southern Mozambique enrolled in a matched case-control study on the etiology of diarrhea," has been formally accepted for publication in PLOS Neglected Tropical Diseases.

Best regards,

Shaden Kamhawi

co-Editor-in-Chief

Paul Brindley

co-Editor-in-Chief
